# The genetic basis of natural variation in a phoretic behavior

Daehan Lee [1,2], Heeseung Yang[1], Jun Kim[1], Shannon Brady[2], Stefan Zdraljevic[2], Mostafa Zamanian[2,3], Heekyeong Kim[4], Young-ki Paik[4,5,6], Leonid Kruglyak[7,8], Erik C. Andersen [2] & Junho Lee[1]

Phoresy is a widespread form of commensalism that facilitates dispersal of one species through an association with a more mobile second species. Dauer larvae of the nematode *Caenorhabditis elegans* exhibit a phoretic behavior called nictation, which could enable interactions with animals such as isopods or snails. Here, we show that natural *C. elegans* isolates differ in nictation. We use quantitative behavioral assays and linkage mapping to identify a genetic locus (*nict-1*) that mediates the phoretic interaction with terrestrial isopods. The *nict-1* locus contains a Piwi-interacting small RNA (piRNA) cluster; we observe that the Piwi Argonaute PRG-1 is involved in the regulation of nictation. Additionally, this locus underlies a trade-off between offspring production and dispersal. Variation in the *nict-1* locus contributes directly to differences in association between nematodes and terrestrial isopods in a laboratory assay. In summary, the piRNA-rich *nict-1* locus could define a novel mechanism underlying phoretic interactions.

[1] Department of Biological Sciences, Institute of Molecular Biology and Genetics, Seoul National University, Seoul 08826, Korea. [2] Department of Molecular Biosciences, Northwestern University, Evanston, IL 60208, USA. [3] Department of Pathobiological Sciences, University of Wisconsin-Madison, Madison, WI 53706, USA. [4] Yonsei Proteome Research Center, Yonsei University, Seoul 03722, Korea. [5] Department of Integrated OMICS for Biomedical Science, Yonsei University, Seoul 03722, Korea. [6] Department of Biochemistry, Yonsei University, Seoul 03722, Korea. [7] Department of Human Genetics and Biological Chemistry, University of California, Los Angeles, CA 90095, USA. [8] Howard Hughes Medical Institute, Chevy Chase, MD 20815, USA. Correspondence and requests for materials should be addressed to E.C.A. (email: erik.andersen@northwestern.edu) or to J.L. (email: elegans@snu.ac.kr)

Species dispersal has been an important topic in evolutionary biology since Charles Darwin's era. He was fascinated by how one species facilitates the migration of another species. For example, he described a phoretic interaction between ducks and freshwater snails after observing just-hatched snails attached to a duck's foot, suggesting a dispersal mechanism for the wide-range distribution of snails[1]. Since that time, numerous reports of phoresy have accumulated[2–4], but the genetic bases of these interactions remain elusive. To address the genetic under-pinnings of a natural phoretic interaction, we investigated the association between the stress-resistant, long-lived dauer larvae[5] of the nematode *Caenorhabditis elegans* and terrestrial isopods[6]. Interspecific association and phoretic dispersal of dauer larvae are conserved among *Caenorhabditis* species[7] (Supplementary Movie 1), and this interaction is facilitated by a dauer-specific phoretic behavior called nictation where dauer larvae lift and wave their bodies, presumably to increase interactions with more mobile species[8, 9] (Supplementary Movie 2).

Because natural variants are known to contribute to phenotypic differences in wild populations, quantitative genetic approaches have been used to successfully elucidate the molecular bases of the natural variation underlying a variety of *C. elegans* traits[10–13]. We investigated natural variation in the nictation behavior of *C. elegans* wild isolates, utilizing a previously established quantitative assay to measure the fraction of nictating dauers among a population of moving dauers[8, 14]. We find significant differences among the strains, including two strains (N2 from Bristol, England and CB4856 from Hawaii, USA) for which recombinant inbred lines have been constructed to enable linkage-mapping approaches[15]. Using this mapping approach, we identified a quantitative trait locus (QTL) on chromosome IV that

contributes to variation in nictation behavior. This locus contains a large cluster of Piwi-interacting small RNAs (piRNAs), which are all regulated by the Piwi Argonaute PRG-1[16]. Because we could not perturb single piRNAs out of many hundreds, we generated *prg-1* mutant strains to test the causal connection of piRNAs to nictation behavior. These results show that the N2 strain contains piRNAs that likely inhibit genes that mediate nictation. We go on to demonstrate that this QTL underlies a phoretic behavior with terrestrial isopods and contributes to a potential fitness trade-off.

## Results

**Natural variation in nictation behavior.** To establish whether natural strains of *C. elegans* vary in a potential dispersal behavior, we focused on nictation—a behavior where the long-lived dauer larvae lift and wave their bodies to increase interactions with larger, more mobile species. We used a previously established quantitative assay to measure the fraction of nictating dauers among a population of moving dauers[8, 14] for a collection of genetically diverse wild strains (Fig. 1a and Supplementary Data 1). We found that 12 strains differed significantly (one-way analysis of variance (ANOVA), $p < 0.001$) in nictation behaviors over a fourfold range, indicating that natural strains likely have diverse phoretic interactions mediated by nictation. We found that a smaller fraction of CB4856 dauers nictate than dauers of the N2 strain (Fig. 1a), and recombinant inbred lines have been constructed to enable linkage-mapping approaches[15] using these two strains. Additionally, N2 dauers show the fifth highest mean nictation fraction out of the 12 strains measured, indicating that differences between these two strains fall within the range of natural behavioral variation.

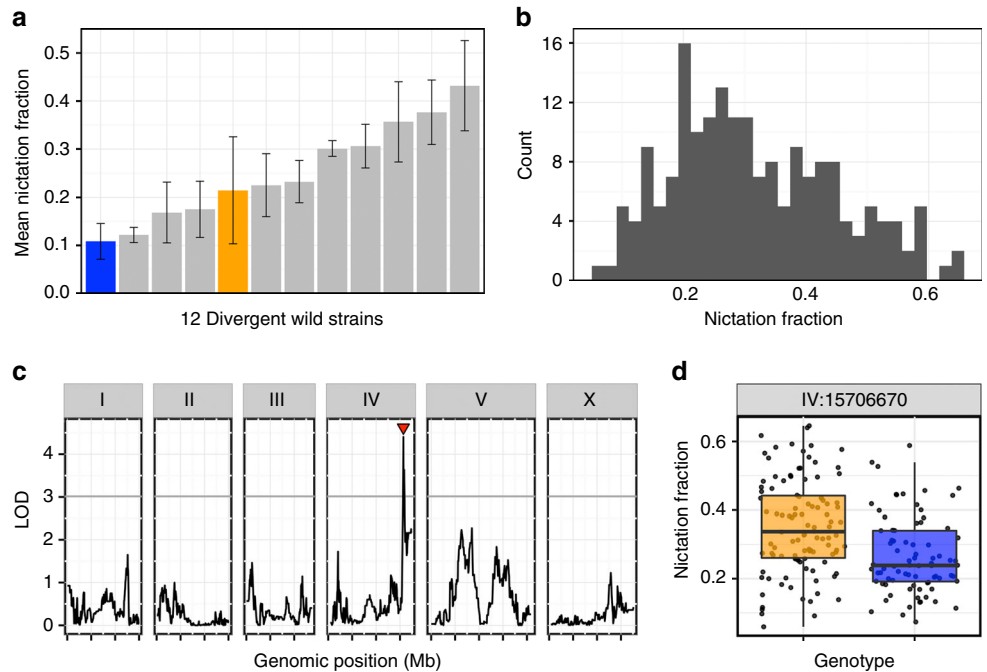

**Fig. 1** *C. elegans* differences in a phoretic behavior map to chromosome IV. **a** The mean nictation fractions of 12 divergent wild isolates are shown (from *left*, CB4856 (*blue*), EG4725, ED3017, JT11398, N2 (*orange*), JU258, JU775, DL238, LKC34, CX11314, MY16, and MY23). Twenty-four replicates of CB4856 and three replicates of all other strains were measured to calculate the means. *Error bars* are standard deviations. **b** A *histogram* of the normalized nictation fractions of 186 N2xCB4856 recombinant inbred advanced intercross lines (RIAILs) is shown. **c** Linkage-mapping results of normalized nictation fraction are shown with genomic position (Mb) on the *x*-axis and logarithm of odds (LOD) score on the *y*-axis. The *tick marks* on the *x*-axis denote every 5 Mb. Each chromosome is in its own box labeled on *top*. The *gray line* is the LOD threshold for 5% genome-wide error rate obtained by permuting the phenotype data and mapping 1000 times. The *red triangle* denotes the peak QTL marker. **d** *Tukey box plots* of N2 (*orange*) and CB4856 (*blue*) are shown with data points plotted behind. The *horizontal line* in the middle of the box is the median, and the *box* denotes the 25th to 75th quantiles of the data. The *vertical line* represents the 1.5 interquartile range

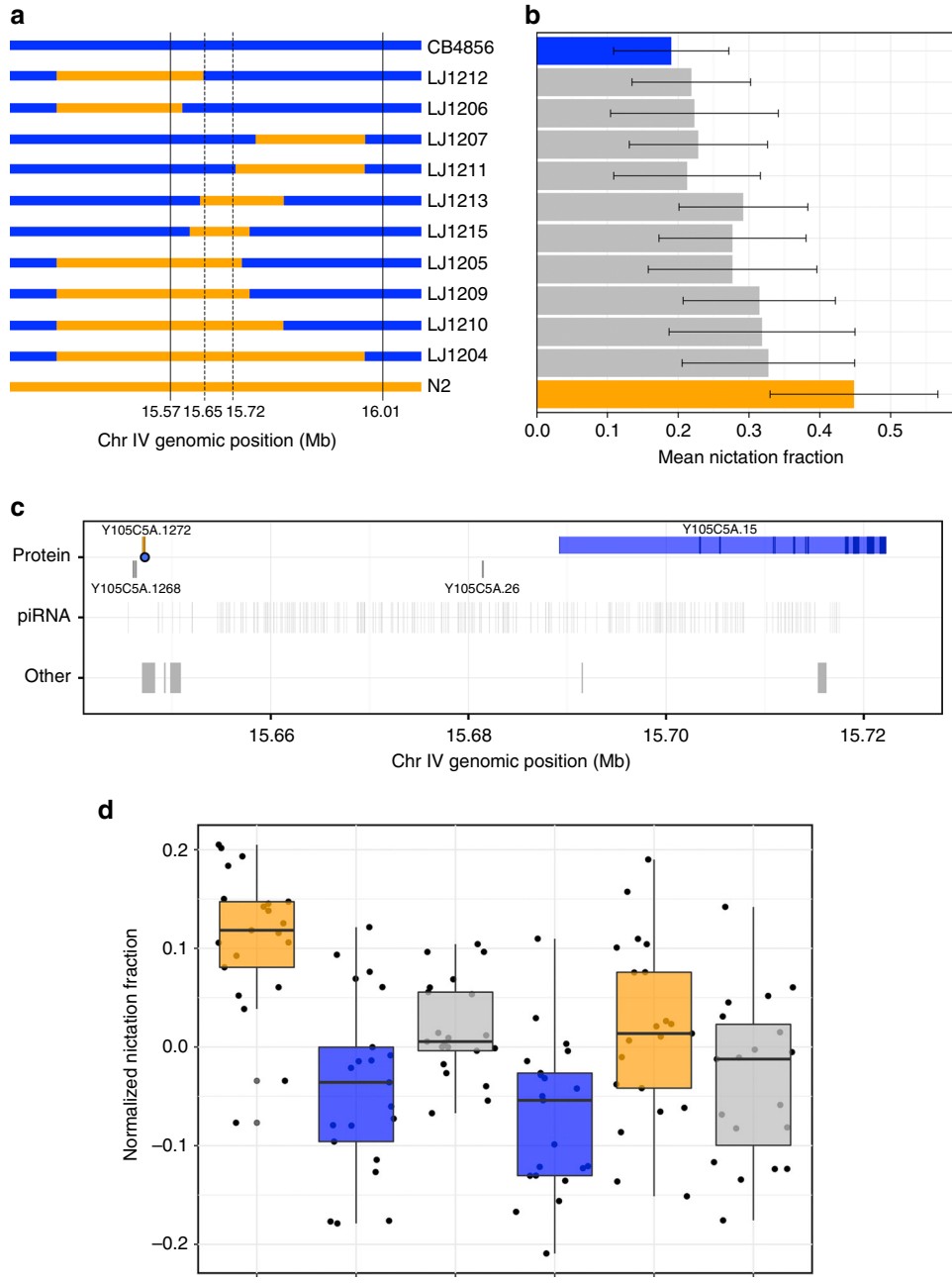

**Fig. 2** Variation in 21U piRNAs underlies nictation differences. **a** Genotypes of near isogenic lines (NILs) represented by *orange* (N2) and *blue* (CB4856) colored bars are shown. Strain labels are to the *right*. *Solid vertical lines* denote the confidence interval from the genetic mapping experiment. *Dotted lines* denote the narrowed genetic interval. **b** The mean nictation fractions of N2 (*orange*), CB4856 (*blue*), and NILs are shown as *horizontal bars*. *Error bars* are standard deviations. The number of biological replicates for each strain range from 17 to 49. **c** Genomic features of NIL-narrowed 73 kb *nict-1* QTL interval are shown. Protein-coding genes (*top*) are colored *orange* (higher expression in the N2 *nict-1* strain), *blue* (higher expression in CB4856 *nict-1* strain), or *gray* (no expression difference between the *nict-1* genotypes). The *blue dot* denotes a predicted functional effect in *Y105C5A.1272*. The locations of piRNAs and other genomics features (tRNAs, snoRNAs, etc.) are shown in *gray* in the *middle* and *bottom rows*. **d** *Tukey box plots* of normalized nictation fractions of N2 (*orange*), LJ1203 (*nict-1* CB4856 > N2, *blue*), ECA586 (*prg-1(ean28)* in the N2 genetic background, *gray*), CB4856 (*blue*), LJ1213 (*nict-1* N2 > CB4856, *orange*), ECA584 (*prg-1(ean30)* in the CB4856 genetic background, *gray*) are shown with data points plotted behind. The *horizontal line* is the median, and the *box* denotes the 25th to 75th quantiles of the data. The *vertical line* represents the 1.5 interquartile range. All strains were scored in 21 biological replicates, except for strain ECA584 where 19 biological replicates were scored. Each biological replicate was composed of three technical replicates

**Identification and fine mapping of the *nict-1* QTL.** We sought the genetic causes underlying this difference in nictation by measuring the behaviors of 186 recombinant inbred advanced intercross lines (RIAILs) derived from a cross between N2 and CB4856 (Fig. 1b and Supplementary Data 2). Linkage mapping revealed a single significant QTL, which we named *nict-1*, on the right arm of chromosome IV (Fig. 1c). This locus does not overlap any of laboratory-derived loci, including *npr-1*, *glp-5*, and *nath-10*, that were discovered previously[17–20]. Recombinant strains with the N2 *nict-1* genotype exhibited a higher nictation ratio compared to those strains with the CB4856 genotype (Fig. 1d), consistent with the parental difference.

To validate and to narrow the *nict-1* QTL, we generated near-isogenic lines (NILs) by crossing the *nict-1* QTL genomic region from N2 into the CB4856 genetic background, as well as performing the reciprocal cross. The phenotypes of these NIL strains confirmed the QTL effect in both genetic backgrounds, with the N2 *nict-1* interval promoting higher nictation than the CB4856 *nict-1* interval (Fig. 2a, b and Supplementary Data 3). By generating recombinants across the confirmed *nict-1* region, we created nine additional NILs containing smaller N2 genomic regions in the CB4856 background. The differences in nictation behaviors of these NILs narrowed the *nict-1* QTL to a 73 kb region (Fig. 2a, b and Supplementary Data 3).

**piRNAs contribute to differences in nictation behavior.** The *nict-1* QTL genomic region contains four protein-coding genes, three pseudogenes, two non-coding RNA genes, and 289 annotated 21U piRNAs genes[21] (Fig. 2c). Using characterized sequence variation between the two parental strains[22–24], we found that only one (*Y105C5A.1272*) of the four protein-coding genes contained variation between the N2 and CB4856 strains. The variant in *Y105C5A.1272* encodes a putative serine-to-glycine change that could alter gene function (Supplementary Data 4). Additionally, we investigated gene expression of the four protein-coding genes and the three putative psuedogenes by quantitative RT-PCR. We found that the N2 strain expresses *Y105C5A.1272* gene, whereas the CB4856 strain does not (Supplementary Data 5). We tested two independently generated deletion alleles of this gene in the N2 *nict-1* genetic background (Supplementary Fig. 2). Although each deletion removes a large proportion of the *Y105C5A.1272* coding sequence, the two mutant strains did not show the same effect on nictation (Supplementary Fig. 1 and Supplementary Data 6), suggesting that *Y105C5A.1272* does not underlie differences in nictation between the two strains. Of the other three protein-coding genes in the interval (Fig. 2c), we found that *Y105C5A.15* had a detectable expression difference between the parental strains and the *nict-1* NILs (Supplementary Data 5).

To test whether this gene expression difference could cause the nictation effect, we assayed a deletion allele of *Y105C5A.15* in the N2 background and found that it does not have an effect on nictation (Supplementary Figs. 1 and 2 and Supplementary Data 6), suggesting that none of the protein-coding genes in the *nict-1* interval play a role in nictation differences between the N2 and CB4856 strains.

Next, we investigated variation in pseudogene sequences and expression in both the N2 and CB4856 strains. We found that the three pseudogenes have variants that would eliminate gene function and remain psuedogenes in both genetic backgrounds. Additionally, expression of these three psuedogenes did not correlate with the differences in the *nict-1* QTL genotype (Supplementary Data 5). These results led us to consider the 289 21U piRNA genes. Because specific perturbations of individual piRNA genes often do not cause loss of gene regulation and the 289 piRNA genes are distributed throughout the 73-kb *nict-1* QTL region, single gene perturbations are not feasible. However, the more than 12,000 21 U piRNA genes, including those genes in this region, require the Piwi Argonaute encoding gene, *prg-1*[17]. Therefore, we globally perturbed piRNA functions by deleting exons one through seven of the *prg-1* gene in both the N2 and CB4856 genetic backgrounds using the CRISPR/Cas9 genome-editing system[25–27]. Loss of *prg-1* in the N2 genetic background resulted in a nictation fraction similar to the CB4856 strain (Fig. 2d). Furthermore, loss of *prg-1* in the CB4856 genetic background had a similar nictation fraction as the CB4856 strain, implicating N2-specific piRNAs in the control of this phoretic behavior. Additionally, an independent *prg-1* loss-of-function allele in the N2 genetic background had a similar effect on nictation (Supplementary Fig. 3 and Supplementary Data 6). These small RNA genes regulate the expression of transposons and also endogenous genes[28]. Any one or multiple of these targeted genes could play a role in the regulation of nictation behavior. Because expression of *prg-1* is restricted to the germline[16], it remains elusive how differences in piRNAs can underlie differences in nictation behaviors.

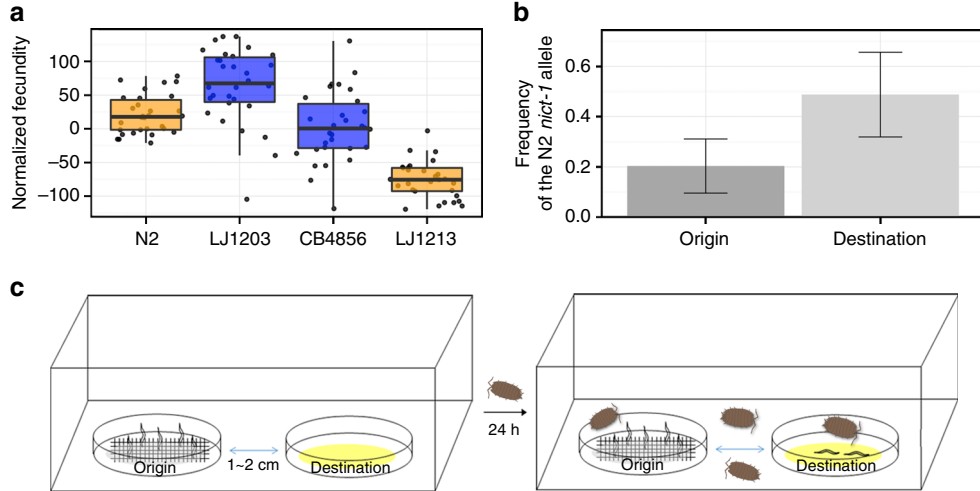

**Fig. 3** The *nict-1* QTL controls a fitness trade-off and species dispersal on isopods. **a** Brood sizes of N2, CB4856, LJ1203 (*nict-1* CB4856 > N2), and LJ1213 (*nict-1* N2 > CB4856) are shown as *Tukey box plots* colored based on the *nict-1* genotype. The *horizontal line* is the median, and the *box* denotes the 25th to 75th quantiles of the data. The *vertical line* represents the 1.5 interquartile range. All four strains were scored in 30 biological replicates, except for LJ1213 where only 29 biological replicates were scored. **b** The mean N2 *nict-1* QTL allele frequencies before (origin) and after (destination) transmission competition assays are plotted. *Error bars* are standard deviations. **c** The experimental scheme of the transmission competition assay is shown. Transmission competition assay chambers contain two nematode culture plates. The "origin" plate contained a mixture of dauer animals from the CB4856 and LJ1213 (*nict-1* N2 > CB4856) strains and was covered with medical gauze to facilitate nictation. The "destination" plate was placed 1–2 cm away with no nematodes on this plate. Terrestrial isopods were added to the chamber and allowed to roam freely for 24 h. Nematodes depend on the isopods for transfer to the "destination" plate from the "origin" plate because no transfer was observed without isopods presence. Eleven different biological replicates were scored

***nict-1* controls a trade-off between nictation and reproduction**.
The *nict-1* QTL likely contributes to different fitness consequences for wild *C. elegans* strains. The NIL strains had brood sizes that were significantly different from their respective parental genetic backgrounds (Fig. 3a and Supplementary Data 7). The offspring production of *nict-1* from N2 crossed into the CB4856 background (LJ1213) was lower than that of the CB4856 parent, and the brood size of *nict-1* from CB4856 crossed into the N2 background (LJ1203) was higher than N2. Therefore, the *nict-1* QTL likely confers a trade-off between nictation and reproduction with the N2 version, promoting nictation but inhibiting offspring production. This result implies that the N2 strain has a functional piRNA that acts to promote nictation and the same or an independent piRNA that acts to inhibit reproduction. Alternatively, a linked N2 variant in a protein-coding gene could exert an opposing effect on reproduction in the N2 *nict-1* genotype. Next, we analyzed a correlation between the nictation fraction and previously determined[22] offspring production of the wild strains (Supplementary Fig. 4). We did not find a strong correlation between these two traits ($r = -0.1666$), suggesting that the trade-off between nictation and reproduction might be specific to the *nict-1* QTL between the N2 and CB4856 strains.

***nict-1* underlies phoretic dispersal mediated by isopods**.
Natural populations of *C. elegans* exhibit local genetic diversity[29], suggesting that intraspecific competition occurs among wild strains in the natural environment. To examine how quantitative variation in nictation behavior relates to the dispersal of *C. elegans* in a competitive environment, we developed a transmission competition assay to quantitatively analyze isopod phoresy (Fig. 3b, c). To mimic this natural phoretic interaction, we used the terrestrial isopod, *Porcellio scaber*, so that we can test whether *C. elegans* can hitchhike to a more favorable environment. Isopods were previously reported as a natural carrier of *C. elegans*[6, 7], and dauers can readily attach to them. We induced dauers from two different *nict-1* genotypes in the same culture to determine whether the N2 *nict-1* genotype, which confers a higher nictation fraction, could confer a higher transmission frequency than the CB4856 *nict-1* genotype. Because of differences in offspring production caused by the trade-off discussed above, the culture contained four times as many dauers of the CB4856 *nict-1* genotype than the N2 *nict-1* genotype (Fig. 3b and Supplementary Data 8). Despite this disparity on the origin plate, we found that dauers with the N2 *nict-1* genotype were transported to the destination plate at a rate four times higher than the rate of the CB4856 *nict-1* genotype. This transport was dependent on the presence of terrestrial isopods. These results suggest that variation in *nict-1* underlies this natural phoretic interaction. Furthermore, given the potential intraspecific competition in the natural environment, our results demonstrate that the *nict-1* QTL could control the ability of strains to colonize new bacteria-rich environments.

## Discussion

We propose that the *nict-1* QTL controls a hitchhiking behavior and phoretic dispersal, as well as a trade-off between dispersal and reproduction, two traits necessary for the survival and evolution of the *C. elegans* species. Behavioral diversity arises from the gene–environment interface where a biological system confronts an ever-changing natural niche, and multiple genes are involved in such dynamic interactions[30]. Although this study identifies a novel difference between the laboratory strain and a wild strain, the observed difference in nictation does not appear to be laboratory-derived because the N2 strain is not exceptional in this behavior, as it is for other traits optimized in the laboratory[31]. Our study provides strong evidence that the nictation behavior is controlled by differences in regulatory small RNAs, which could act via a large number of distributed effects to create robust environmental adaptations[32]. Involvement of the piRNA pathway in neuronal plasticity is suggested from studies of the *Aplysia* central nervous system and the mammalian brain, implying conserved regulatory roles among metazoans[33–36]. Studies of a large collection of diverse wild strains are required to understand the broad applicability of the piRNA-rich *nict-1* QTL effect across the *C. elegans* population. This proposed molecular mechanism of behavioral control enables future studies of this phoretic interaction and its precise genetic causes across *C. elegans* and related nematode species.

## Methods

**Strains**. *Wild isolates*. CB4856, CX11314, DL238, ED3017, EG4725, JT11398, JU258, JU775, LKC34, MY16, MY23, and N2. Strain data including isolation location, isotype information, and more are available from CeNDR (https://www.elegansvariation.org)[37].

*RIAILs*. QX2-QX17, QX19, QX20, QX22, QX24-QX27, QX29, QX32-QX34, QX36-QX45, QX47-QX49, QX51-QX88 QX90, QX92-QX97, QX99, QX100, QX102-QX104, QX106-QX118, QX120-QX122, QX124, QX125, QX127-QX129, QX131, QX134, QX137, QX138, QX140, QX147, QX148, QX153, QX154, QX156-QX166, QX169-QX178, QX180-QX193, QX195-QX198, QX200-QX207, QX212, QX213, QX216-QX221, QX224-QX227, QX229- QX237, and QX239.

*NILs*. LJ1203 *snuIR3 (nict-1 QTL, CB4856 > N2)*, LJ1204 *snuIR4 (nict-1 QTL, N2 > CB4856)*, LJ1205 *snuIR5 (nict-1 QTL, N2 > CB4856)*, LJ1206 *snuIR6 (nict-1 QTL, N2 > CB4856)*, LJ1207 *snuIR7 (nict-1 QTL, N2 > CB4856)*, LJ1209 *snuIR9 (nict-1 QTL, N2 > CB4856)*, LJ1210 *snuIR10 (nict-1 QTL, N2 > CB4856)*, LJ1211 *snuIR11 (nict-1 QTL, N2 > CB4856)*, LJ1212 *snuIR12 (nict-1 QTL, N2 > CB4856)*, LJ1213 *snuIR13 (nict-1 QTL, N2 > CB4856)*, LJ1215 *snuIR15 (nict-1 QTL, N2 > CB4856)*.

*Mutants*. FX6979 *Y105C5A.1272(tm6979)*, FX1161 *Y105C5A.15(tm1161)*, ECA286 *prg-1(n4357)* (backcrossed to the Andersen lab N2 10 times), ECA584 *prg-1(ean28)*, ECA586 *prg-1(ean30)*, LJ1216 *snuIR13 (nict-1 QTL, N2 > CB4856), Y105C5A.1272(snu33)*.

**Dauer induction and nictation assays**. Ten to twenty L4 larvae or young adults were transferred to synthetic pheromone plates containing agar (10 g/L), agarose (7 g/L), NaCl (2 g/L), $KH_2PO_4$ (3 g/L), $K_2HPO_4$ (0.5 g/L), cholesterol (8 mg/L), and synthetic pheromone-ascaroside 1, 2, 3 (2 mg/L each)[38, 39] seeded with *Escherichia coli* OP50 at 25 °C for dauer induction[8, 14]. After 4 days, dauers were morphologically identified by their dark intestines and radially constricted bodies. Micro-dirt chips were made by pouring 3.5% agar solution onto poly-dimethylsiloxane (PDMS) mold[14]. Solidified agar micro-dirt chip was detached from the PDMS mold and dried for 90 min at 37 °C. More than 30 dauers were collected by glass capillary (Kimble Chase Life Science and Research Products LLC.) using M9 buffer and mounted on a micro-dirt chip. After 10–30 min, when dauers were actively moving, a fraction of nictating dauers among moving dauers on a micro-dirt chip was measured three times consecutively, and the mean value of three technical replicates was represented as the nictation fraction for that biological replicate. The mean nictation fraction was calculated from the independent biological replicates of multiple nictation assays. We calculated the normalized nictation value by fitting a linear model, nictation fraction ~ assay date. Number of independent biological replicates for each assay is described in figure legends.

**Quantitative genetic analyses**. The nictation fractions of 186 RIAIL strains from an advanced intercross between N2 and CB4856 were scored as described above. The phenotype data and genotype data were entered into R and scaled to have a mean of zero and a variance of one for linkage analysis. QTLs were detected by calculating logarithm of odds (LOD) scores for each marker and each trait as $-n(\ln(1-r^2)/2\ln(10))$, where $r$ is the Pearson correlation coefficient between RIAIL genotypes at the marker and phenotype trait values[40, 41]. We randomly permuted the phenotype values of each RIAIL while maintaining correlation structure among phenotypes 1000 times to estimate significance empirically. Confidence intervals were defined as the regions contained within a 1.5 LOD drop from the maximum LOD score.

**NIL construction**. LJ1203 *snuIR3 (nict-1 QTL, CB4856 > N2)* was made from QX4, a RIAIL with the CB4856 *nict-1* QTL interval on an otherwise N2 chromosome IV. QX4 was backcrossed to the N2 parent six times while selecting for the CB4856 *nict-1* QTL by single-nucleotide polymorphism (SNP) genotyping. LJ1204 *snuIR4 (nict-1 QTL, N2 > CB4856)* was made from QX162, a RIAIL with the N2 *nict-1* QTL interval in an otherwise CB4856 chromosome IV. QX162 was backcrossed to

the CB4856 parent six times while selecting for the N2 nict-1 QTL by SNP genotyping. To genotype the nict-1 QTL interval determined by linkage mapping (15.57–16.01 Mb), we used a DraI restriction fragment length polymorphism (RFLP) to genotype a SNP at 15,570,359 (left, marker haw64768) and 16,004,711 (right, marker haw65532)[42]. Other NILs were generated by backcrossing LJ1204 to CB4856 further. Precise introgression breakpoints of each NIL were determined by genotyping SNP markers at position described in Supplementary Table 1 using the primers described in Supplementary Table 2.

**Analysis of genes in the QTL interval.** C. elegans protein-coding and non-coding RNA genes were extracted using a custom Python script (Wormbase GFF version WS248). Variants within the QTL interval that distinguish N2 from CB4856 and that are predicted to have "moderate" or "high" functional effects were identified using the R cegwas package[37, 43] and were plotted with respect to gene models.

**CRISPR/Cas9 site-directed mutagenesis.** For mutating Y105C5A.1272, purified Cas9 protein and two Y105C5A.1272 targeting single-guide RNAs (sgRNAs; targeting sequence 5′-CATTTATGACGAGAACATTGAGG-3′ and 5′-GTCGAT GCGAAGTACTTCCCTGG-3′) were purchased from Toolgen Inc. Cas9 protein was resuspended with 20 mM HEPES-KOH, pH 7.5, 150 mM KCl, 1 mM dithiothreitol, and 10% glycerol to 8 μg/μL. Each sgRNA was resuspended with RNase-free D.W to 5 μg/μL. Then, Cas9 protein and two sgRNA mixtures were mixed at 1:1 volume ratio with sur-5::GFP injection marker (100 ng/μL) and NEBuffer 3 (1×). After incubating the mixture at 37 °C for 10 min, the RNP complex was directly injected into a gonad of LJ1213 young adult animals. Green fluorescent protein-positive F1 progeny were singled out and subjected to single-animal lysis after laying enough embryos to propagate the strain. Deletions were detected by single-animal PCR using 5′-AGAGGAGAATGCACGAGGAA-3′ and 5′-TGCTT CCTCATCTACCAGCA-3′ primers. F2 homozygotes were segregated by using same method.

Mutations in prg-1 were generated from the following procedure: purified Cas9 protein and two prg-1 targeting crRNAs (targeting sequence 5′-GTTAGCCTT CGAATCAACGG-3′ and 5′-CGCTGTGACCGACAAAGC-3′) were purchased from Integrated DNA Technology Inc. We used dpy-10 co-CRISPR marker crRNA (targeting sequence 5′-GCTACCATAGGCACCACGAG-3′) and repair template (5′-CACTTGAACTTCAATACGGCAAGATGAGAATGACTGGAAACCGTACC GCATGCGGTGCCTATGGTAGCGGAGCTTCACATGGCTTCAGACCAACA GCCTAT-3′) to increase the yield of the prg-1 mutation. Cas9 protein was resuspended with 20 mM HEPES pH 8.0, 500 mM KCl, 20% glycerol to 11 mg/mL (69 μM). A volume of 0.34 μL of tracrRNA (200 μM), 0.20 μL of dpy-10 crRNA (100 μM), and 0.48 μL of each prg-1 crRNAs (100 μM) 1.75 μL were mixed and incubated at 95 °C for 5 min and 10 °C for 5 min. Cas9 protein was added and incubated for 5 min at 25 °C. A volume of 0.67 μL of dpy-10 repair oligo (10 μM) and nuclease-free water was added to total 10 μL volume. The mixture was injected into gonads of N2 and CB4856 young adult animals. Rol-positive F1 progeny were singled out and subjected to single-animal lysis and PCR for detecting deletion using 5′-GGGTACTATCCAACCCGATCTTTTCATTCG-3′ and 5′-CGCGTTT CGTGACAATGATAAATGCG-3′. F2 homozygotes were segregated by using same method.

**Quantitative RT-PCR.** Mixed-stage animals cultured at 25 °C on NGM lite (2.0 g NaCl, 4.0 g Bactotryptone, 3.0 g $KH_2PO_4$, 0.5 g $K_2HPO_4$, 8 mg Cholesterol, and 20 g Agar in 1 L of water) were collected. RNA was isolated by Trizol treatment and reverse-transcribed by TOPscript cDNA Synthesis kit (Enzynomix Inc.). RNA from six independent biological replicates was analyzed. Quantitative RT-PCR was performed using iQ™ SYBR® Green Supermix (Bio-Rad Laboratories Inc.).

**Brood size assays.** Twenty young adults of each strain were transferred on 10 cm NGM plates seeded with OP50 and allowed to lay embryos. After 1–2 h, adult animals were removed and fewer than 100 embryos were grown at 20 °C. After 2–3 days, 10 L4 larvae or young adults were singled and transferred to 55 mm NGM lite agar plates[44] seeded with OP50 at 25 °C. Animals were transferred to new plates every 24 h for 3 days. The number of offspring on each plate was counted manually after 2 days from transfer. Normalized fecundity was calculated using a linear model, brood size ~ assay date.

**Transmission competition assays.** Ten synchronized young adults from CB4856 and LJ1213 (nict-1 N2 > CB4856) were placed together on a single pheromone plate, where dauers were induced by the same method as described above. After 4 days, when hundreds of dauers were observed, more than 20 dauers were singly collected by glass capillary and genotyped at the haw64888 SNP (IV, 15,696,457) in the nict-1 QTL region using single-animal lysis and a PCR-based DraI RFLP assay[42]. After that step, medical gauze was mounted on the pheromone plate as a platform to enable nictation of the remaining dauer larvae. The plate with nictating dauers and a new plate with OP50 food were placed 1–2 cm apart within a small plastic box. Four to six Porcellio scaber terrestrial isopods, collected from Seoul National University campus, were also placed in the box. The isopods moved freely for 24 h. We genotyped the transferred animals at the haw64888 SNP (IV, 15,696,457) in the nict-1 QTL using the same method as described above.

**Data availability**. All data are available in Supplementary Data 1–8.

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

## Acknowledgements

We thank M.-A. Félix for experimental advice on the transmission competition assay along with D.O. and S.K. for providing the micro-dirt chip PDMS mold. We thank the National BioResource Project (NBRP) and the *Caenorhabditis* Genetics Center, which is funded by the NIH Office of Research Infrastructure Programs (P40 OD010440) for some strains. This work was supported by Samsung Science and Technology Foundation under Project Number SSTF-BA1501-04. D.L. was supported by POSCO Science Fellowship from the POSCO TJ Park Foundation. E.C.A. was supported by a grant from the National Institutes of Health R01 subcontract (GM107227), the Chicago Biomedical Consortium with support from the Searle Funds at the Chicago Community Trust, and an American Cancer Society Research Scholar grant (127313-RSG-15-135-01-DD) along with support from the Cell and Molecular Basis of Disease training grant (T32GM008061) to S.Z. and Biotechnology training grant (T32GM008449) to S.C.B.

## Author contributions

D.L., E.C.A., and J.L. designed experiments, analyzed and interpreted the data, and wrote the manuscript. H.Y. performed the nictation assay for 12 wild isolates and J.K. performed the brood size assay. S.C.B. and S.Z. generated the *prg-1* CRISPR/cas9 deletion alleles. M.Z. analyzed genome variation and backcrossed the *prg-1(n4357)* allele. H.K. and Y.P. supplied synthetic pheromones. L.K. edited the manuscript.

## Additional information

**Competing interests:** The authors declare no competing financial interests.

