## [Peer Review File · Nature Communications]

Reviewers' comments:

Reviewer #1 (Remarks to the Author):

Genetic basis of natural variation in a phoretic behavior
Lee et al

This is an interesting but preliminary paper describing the mapping of a difference between N2 and CB4856 *C. elegans* in their nictation behavior. It is interesting, but the methodology is poorly explained - in particular experimental sizes - and this detracts from confidence in the results.

The association with piRNAs is very intriguing, and the authors provide preliminary support for an effect of knocking out piRNA processing in N2 on nictation, but given the measured fecundity effect of the "nict-1" locus and the roles of piRNAs in genome defense, it may be that they are measuring effects on general viability.

RNAi of *prg-1* in CB strains, or even in other of the strains the authors screened in their fig1a, would be informative in this.

Abstract

Very short and does not announce main findings

40 "facultative phoretic interaction between the nematode *Caenorhabditis elegans* and terrestrial isopods" I hope there is some data on interaction...

text

47 "but the genetic basis" - bases, unless the authors believe that all phoresy involves the same genetics!

63 "We found that a smaller fraction of CB4856 dauers nictate than dauers of the N2 strain" - do the authors think this is "natural" variation, or variation consequent to the maintenance of (N@ especially) in lab culture; N2 is known to be mutant at loci implicated in a wide range of behaviours.

80 "four protein-coding genes, three pseudogenes" - tested are
Y105C5A.1272 (variant but deletion has no effect)

Y105C5A.15 (expression difference in bulk mRNA but deletion has no effect)

?? two others?

?? are pseudogenes non-pseudogenised in CB4856?

88 "did not show same effect" - did not show THE same effect

104 "loss of *prg-1* function in the N2 genetic background" - how is CB4856 nictation affected by *prg-1* ablation? This may be a general effect of loss of *prg-1* on viability

113 "The brood size of nict-1" - this suggests a general issue of fertility associated with nict-1, which could correlate with the piRNA processing issue; was this explored?

118 "local genetic diversit" - DIVERSITY

122 "we introduced terrestrial isopods" - which species?

methods

These are generally poorly described. In particular the number of independent trials for each assay are not given - are the bars on the figures SDs? This makes it very hard to

assess the significance of the differences plotted. "Normalized nictation percent" is not explained.

151 "After 10 to 30 min" - the methodology is rather inexact - did the proportion change with time (ie is 10 min == 30 min?). Is the effect size such that assay of only 10 animals is sufficient? The Mean nictation fraction" measures in Figures 1 and 2 differ for N2 and CB strains (N2 0.22 vs 0.45, CB 0.1 vs 0.2) - this suggests that the assay shows a lot of variation between trials...

154 "Quantitative genetic analysis" this paragraph is poorly formatted

174 "and subjected to genotyping SNP at 15,696,457 (haw64888) in nict-1 QTL" - not clear what this means... a single SNP was typed?

169 "Transmission competition assays" - given that the authors have identified a fecundity difference between the two strains, is it unsurprising that more N2 are carried when they likely dominate the culture? Its not clear how this was controlled. Its not clear how the data for figure 3b was calculated - number of trials, etc.

Reviewer #2 (Remarks to the Author):

This is a nice QTL study of nictation, a complex behavioral trait in *C. elegans*, which includes a simple ecological connection through quantification of phoretic interaction. This is a technically solid report and I appreciate the behaviorally and ecologically integrative approach. My recommendation is for publication after considering the below comments:

1) Error bars need to be defined in figures.

2) Fig 3a. How is the normalized fecundity calculated? What is plotted isn't clear to me from the text or figure legend.

Minor:

3) line 118. Should be "diversity"

4) Fig2d. Y-axis label should be "percentage" and not "percent"

5) Methods, line 177. Maybe 1 to 2 cm "spacing" would be more clearer than "interval"

Reviewer #3 (Remarks to the Author):

The paper by Lee et al. describes the identification of a genetic locus by which modulates *C. elegans* nictation and phoretic behavior. The authors used recombinant inbred lines, generated near isogenic lines and narrowed the nictation locus to a 73 kb region on chromosome 4. Since the mutation in *prg-1* elicits the phenotype that resembles Hawaiian polymorphism, the authors conclude that nictation is controlled by non-coding piRNAs. The paper uses genetics to elucidate the underlying mechanism of nictation and phoretic

behavior in *C. elegans*.

This is a well-written paper with interesting conclusions. The following are suggestions to the authors. Perhaps the authors will address these points to enhance the paper.

Major suggestion

(1) The authors previously found that nictation is mediated by acetylcholine signaling in the nervous system. Does the 73 kb *nict-1* locus modulate *cha-1* expression or acetylcholine production?

(2) Along the same line, do piRNAs act in cholinergic neurons? Perhaps doing a rescue experiment of *prg-1* in IL2 neurons?

Minor suggestion

(1) Do *prg-1* mutant and the 73 kb *nict-1* locus influence IL2 neural activity?

Response to reviewers' comments

Thank you very much for the critical assessment of our manuscript. We believe that the manuscript is vastly improved based on the suggested edits and experiments. The reviewers' comments are in bold, and our responses are in non-bold.

Reviewer #1 (Remarks to the Author):

This is an interesting but preliminary paper describing the mapping of difference between N2 and CB4856 *C. elegans* in their nictation behavior. It is interesting, but the methodology is poorly explained in particular experimental sizes - and this detracts from confidence in the results.

Our original submission was transferred from *Nature Ecology and Evolution* and written as a Brief Communication. This format led to shortened explanations for many points. We apologize for the lack of sufficient explanation on methodology. Because we are free from the previous word limits, we supplemented the explanations of the assay, sample sizes, and statistics in the manuscript. We appreciate your advice on how to strengthen our arguments. To directly address the sample size issues in this response, we assayed nictation from three technical replicates of over 30 animals in each replicate with a large number of independent biological replicates. We added the number of biological replicates to the figure legends. Also, the raw data for each biological replicate is in the Supplementary information. For example, we performed 17 to 49 independent biological replicates for the fine mapping of the *nict-1* QTL and *prg-1* effect (Figure 2), which means we measured nictation for at least 1500 animals from every strain in every assay. This level of replication gives us the statistical power to detect these behavioral differences, but the assays are onerous because of this high level of replication.

The association with piRNAs is very intriguing, and the authors provide preliminary support for an effect of knocking out piRNA processing in N2 on nictation, but given the measured fecundity effect of the "nict-1" locus and the roles of piRNAs in genome defense, it may be that they are measuring effects on general viability. RNAi of *prg-1* in CB strains, or even in other of the strains the authors screened in their fig1a, would be informative in this.

Thank you for this concern. We appreciate your suggestions given the diverse roles that piRNAs likely have in *C. elegans* biology. We analyzed a correlation between nictation ratio and offspring production for the wild strains analyzed in Figure 1A. These new data are included in the supplement as Figure S4. We see no correlation between these two traits, indicating that nictation ratios are not reflective of general viability.

Additionally, we went further to investigate the role of *prg-1* in the nictation phenotype. It has been reported previously the CB4856 wild strain (and other wild strains) have differential responses to RNAi (Rockman and Kammenga labs). For this reason, RNAi is not an effective means to test gene function across wild *C. elegans* strains. We used CRISPR/Cas9 genome editing to generate an identical *prg-1* deletion allele in both the N2 and CB4856 strain backgrounds. We scored these strains (and the parents) for nictation effects and found that they recapitulated the effects we observed using introgressed strains previously. These results definitively show that the effect on nictation by this QTL is mediated by piRNAs and *prg-1*. The new figure is Figure 2d. The *prg-1* knockout in the CB4856 genetic background has the same nictation ratio as the CB4856 strain, suggesting that an N2 specific piRNA mediates nictation. Additionally, these results show that general strain sickness is not the underlying cause of nictation differences.

Abstract

Very short and does not announce main findings

As stated previously, the Brief Communication format led to a short abstract. We expanded the abstract to address and explain our major findings.

40 "facultative phoretic interaction between the nematode *Caenorhabditis elegans* and terrestrial isopods" I hope there is some data on interaction...

We are not sure what the reviewer meant in this comment. If the reviewer thought that a reference delineating interactions between *C. elegans* and isopods would strengthen our case, we provided the Frezal and Felix *eLife* reference to explain that this interaction has been identified previously. Additionally, our data from the transmission competition assay (Figure 3) clearly demonstrated that the interspecific phoretic interaction differs between strains with different piRNA-rich *nict-1* QTL alleles.

text

47 "but the genetic basis" - bases, unless the authors believe that all phoresy involves the same genetics!

Corrected.

63 "We found that a smaller fraction of CB4856 dauers nictate than dauers of the N2 strain" _ do the authors think this is "natural" variation, or variation consequent to the maintenance of (N2 especially) in lab culture; N2 is known to be mutant at loci implicated in a wide range of behaviours.

Thank you for addressing this important point. To test the hypothesis of laboratory derivation vs. a natural allele, we must identify the specific variant and then look at the population-wide allele frequency to test this hypothesis. If N2 is the only strain with this variant, then either it is a rare natural variant only in that lineage or it is laboratory derived. Because no frozen samples of the N2 strain exist before laboratory propagation, no definitive way exists to differentiate these two possibilities. If the allele is found in many natural strains, then it is a natural allele. Please see a previous review we published (Sterken *et al. Trends in Genetics* 2015) for further explanations. For these reasons, it is formally possible that the variation in nictation is caused by a laboratory-derived allele, but we do not believe it to be the case for the following reasons. (1) Laboratory-derived variation in the neuropeptide receptor *npr-1* underlies differences in most behaviors between the N2 and CB4856 strains. Our QTL on chromosome IV does not overlap the location of *npr-1* on chromosome X, so it likely plays no role in nictation. Additionally, we explicitly tested strains with differences in *npr-1*. Surprisingly, we found that the CB4856 version of *npr-1* had a higher nictation ratio than the N2 version of *npr-1*, suggesting the opposite result in that laboratory culture might decrease the performance of nictation behavior of the N2 strain. We believe this result argues that the nictation difference between N2 and CB4856 did not originate in the lab, and if it did, it went in the opposite direction than our QTL analysis showed. These data are a part of an upcoming study, which we do not want to include in this manuscript. We mentioned the lack of overlap between the *nict-1* QTL and *npr-1* (and other laboratory-derived loci *glb-5* and *nath-10*) in the final section of the manuscript to support our belief that laboratory adaptation does not explain the differences in nictation. (2) The range of nictation differences observed across the wild strains (Figure 1A) includes N2, but N2 is not an outlier. It is only the fifth highest nictation ratio out of 12 strains measured. For all of the laboratory-derived behaviors

investigated so far, N2 is the most extreme strain because it performs better in laboratory conditions. We added a section to the final section of the manuscript to outline the difficulty in making this assessment of natural vs. laboratory-derived variation.

80 "four protein-coding genes, three pseudogenes" - tested are Y105C5A.1272 (variant but deletion has no effect) Y105C5A.15 (expression difference in bulk mRNA but deletion has no effect)

?? two others?

?? are pseudogenes non-pseudogenised in CB4856?

We apologize for this confusion. The region contains four protein-coding genes. Only two of those four genes have a change in the amino acid sequence between the N2 and CB4856 strains. The other two genes have no amino acid changes and no differences in gene expression, so they are not candidate genes in this trait. We edited the text to make this point more clear.

The three pseudogenes are pseudogenes in the N2 strain. We investigated their structure in the CB4856 strain and found that each pseudogene has variants that would eliminate function. Additionally, the three pseudogenes have no expression differences that correlate with the *nict-1* genotype. Therefore, they are likely ancient pseudogenes shared by both the N2 and CB4856 strains. Thank you for this suggestion. We edited the text to add these new results.

88 "did not show same effect" - did not show THE same effect

Corrected.

104 "loss of prg-1 function in the N2 genetic background" - how is CB4856 nictation affected by prg-1 ablation? This may be a general effect of loss of prg-1 on viability

As we described above, we have ablated *prg-1* function in both the N2 and CB4856 strains using CRISPR/cas9 genome editing that created the same deletion *prg-1* coding sequence in both genetic backgrounds. These perturbations do not differentially affect viability of the two mutant strains. Additionally, as we pointed out above, we see no correlation between fitness traits (viability and offspring production) and nictation ratios. We tested nictation ratios after loss of *prg-1* in both strain backgrounds with the new genome-edited strains. These data replaced the existing data in Figure 2d. These new conclusions support the previous results that *prg-1* and piRNA function are required in the N2 strain to cause differences in nictation. The CB4856 strain lacks this function. We edited the text to make these points more clear.

113 "The brood size of nict-1" - this suggests a general issue of fertility associated with nict-1, which could correlate with the piRNA processing issue; was this explored?

The specific point in this comment is possibly a misunderstanding of our results. We see a trade-off between nictation ratio and fertility mediated by the *nict-1* QTL region. The N2 version of *nict-1* causes higher nictation but lower fertility, suggesting that the same or tightly linked functional genetic factor(s) exert opposite effects on nictation and reproduction. Reciprocally, the CB4856 version of the *nict-1* QTL causes decreased nictation but increased fertility. The N2 and CB4856 *prg-1* mutant strains both have both lower fertility but only N2 has lower nictation. It is important to note that the piRNAs between the N2 and CB4856 strains are not identical. Therefore, this result indicates that the N2 strain has a functional piRNA that acts to promote nictation and the same or an independent piRNA that acts to inhibit reproduction. The CB4856

strain lacks these piRNA activities as shown by the *prg-1* mutant strain in that genetic background. We edited the text to make these points more clearly.

118 "local genetic diversit" - DIVERSITY

Corrected.

122 "we introduced terrestrial isopods" - which species?

We used the *Porcellio scaber* species and have noted it in the text.

methods

These are generally poorly described. In particular the number of independent trials for each assay are not given - are the bars on the figures SDs? This makes it very hard to assess the significance of the differences plotted. "Normalized nictation percent" is not explained.

Again, because of the Brief Communication format, we were limited to fewer than 500 words in the Methods. We are sorry for the lack of sufficient descriptions. We added details and believe we have addressed this concern in the revised version. For the normalization point specifically, we calculated the normalized trait by fitting a linear model, nictation fraction \sim assay date. We explained the statistics in the updated methods.

151 "After 10 to 30 min" - the methodology is rather inexact - did the proportion change with time (ie is 10 min == 30 min?). Is the effect size such that assay of only 10 animals is sufficient? The Mean nictation fraction" measures in Figures 1 and 2 differ for N2 and CB strains (N2 0.22 vs 0.45, CB 0.1 vs 0.2) - this suggests that the assay shows a lot of variation between trials...

We apologize for the confusion. When dauers are released from the glass capillary filled with M9 solution onto micro-dirt chip, they are lethargic for 10 to 30 minutes. The length of lethargy varies among trials. However, after most of dauers become active, the proportion of nictating animals does not vary. As described, more than 30 dauers were placed on micro-dirt chip and at least 20 of them were scored. As described above, we scored nictation from three technical replicates in dozens of independent biological replicates. These details were added to the methods section.

We believe that variability among traits is caused by environmental factors, including temperature and humidity. We scored nictation ratios in temperature and humidity controlled conditions, but we could still observe variation among trials. Therefore, other factors could cause these differences in nictation among and within trials. However, as explained in the manuscript, the differences between or among the strains were statistically significant.

154 "Quantitative genetic analysis" this paragraph is poorly formatted

We are not sure what the reviewer means about formatting. We provided additional details about the quantitative genetic mapping experiments to address this concern.

174 "and subjected to genotyping SNP at 15,696,457 (haw64888) in nict-1 QTL" - not clear what this means... a single SNP was typed?

Yes, a single SNP at position 15,696,457 on Chromosome IV was genotyped. We added details to the methods about how this SNP was genotyped using PCR and a *DraI* RFLP.

169 "Transmission competition assays" - given that the authors have identified a fecundity difference between the two strains, is it unsurprising that more N2 are carried when they likely dominate the culture? Its not clear how this was controlled. Its not clear how the data for figure 3b was calculated - number of trials, etc.

We tested 11 independent trials for this experiment (Supplementary Table 8), which gave us consistent results among trials. As we described in figure legend, the dauers of CB4856 and LJ1213 (*nict-1* N2>CB4856) are competing in the assay. In Figure 3b, we show that only one fifth of dauers on the starting competition plate were LJ1213 (N2 *nict-1*), which is consistent with lower fecundity of this strain. Because of this disparity, our results are even more striking. The strain with the lower fecundity and representation on the "origin" plate (LJ1213) was transferred five times more readily to the "destination" plate than the CB4856 strain. We added more detail to the experimental description and the results to help clarify this exciting result.

Reviewer #2 (Remarks to the Author):

This is a nice QTL study of nictation, a complex behavioral trait in *C. elegans*, which includes a simple ecological connection through quantification of phoretic interaction. This is a technically solid report and I appreciate the behaviorally and ecologically integrative approach. My recommendation is for publication after considering the below comments:

1) Error bars need to be defined in figures.

The figures contain bar plots or box plots. We explained the bar plots contain standard deviations. The Tukey box plots show the range from 25th to 75th quartiles. These details were added to the figure descriptions.

2) Fig 3a. How is the normalized fecundity calculated? What is plotted isn't clear to me from the text or figure legend.

As explained above, the normalized fecundity was calculated using a linear model (brood size ~ assay date). Those regressed values were plotted. We explained more in the methods and figure legends to clarify this point.

Minor:

3) line 118. Should be "diversity"

Corrected.

4) Fig2d. Y-axis label should be "percentage" and not "percent"

Corrected.

5) Methods, line 177. Maybe 1 to 2 cm "spacing" would be more clearer than "interval"

Corrected.

Reviewer #3 (Remarks to the Author):

The paper by Lee et al. describes the identification of a genetic locus by which modulates *C. elegans* nictation and phoretic behavior. The authors used recombinant inbred lines, generated near isogenic lines and narrowed the nictation locus to a 73 kb region on chromosome 4. Since the mutation in *prg-1* elicits the phenotype that resembles Hawaiian polymorphism, the authors conclude that nictation is controlled by non-coding piRNAs. The paper uses genetics to elucidate the underlying mechanism of nictation and phoretic behavior in *C. elegans*.

This is a well-written paper with interesting conclusions. The following are suggestions to the authors. Perhaps the authors will address these points to enhance the paper.

Major suggestion

(1) The authors previously found that nictation is mediated by acetylcholine signaling in the nervous system. Does the 73 kb *nict-1* locus modulate *cha-1* expression or acetylcholine production?

Thank you for this suggestion. As you suggested, the underlying neural mechanism for nictation variation can be explained by differences in the acetylcholine signaling. We tested the hypothesis that increased nictation in N2 *nict-1* genotype might be correlated with increased acetylcholine signaling. To test this hypothesis, we performed qRT-PCR experiments for *cha-1* expression in dauer larvae of the CB4856 and LJ1213 (*nict-1*, N2>CB4856) strains, which we used for both nictation assay and transmission competition assay where LJ1213 strain with N2 *nict-1* QTL shows higher frequency for both nictation and phoretic dispersal. As a result, we observed the decreased *cha-1* expression in nictation-enhanced LJ1213 (*nict-1*, N2>CB4856) strain, which is an opposite result from the hypothesis, indicating differences in acetylcholine signaling is not a causal mechanism for *nict-1* QTL effect.

Figure. *cha-1* expression analysis on different *nict-1* genotype strains

(2) Along the same line, do piRNAs act in cholinergic neurons? Perhaps doing a rescue experiment of *prg-1* in IL2 neurons?

This suggestion is an interesting extension that we did not consider initially. It is known that the expression of PRG-1 is restricted to germline (Batista et al. *Molecular Cell* 2008) so expressing

prg-1 in IL2 neurons would be an exogenous effect. We added more description to the end of the manuscript to address this point specifically. In general, piRNA function in physiology and behavior has not been explored, and it would be exciting to investigate how variation in piRNAs ultimately causes behavioral variation in future studies.

In addition, from our recently published paper (Lee *et al.* BBRC 2017), the regulatory pathway involved in nictation does not necessarily act in IL2 neurons in the dauer stage. We identified that insulin signaling acts in amphid neurons and TGF-beta signaling acts in the RIM/RIC interneuron. The *nict-1* locus might also regulate nictation via any of these signaling pathways or unknown regulatory mechanism in different cells other than the IL2 neurons. We investigated amphid and IL2 neurons by dye filling of both LJ1203 (CB4856>N2 NIL) and *prg-1* animals and found no obvious defects in morphology (below).

Fig. Dil staining of sensory neurons including IL2 and amphid neurons. (left) LJ1203 (*nict-1*, CB4856>N2) (right) ECA586 *prg-1*(*ean30*)

Minor suggestion

(1) Do *prg-1* mutant and the 73 kb *nict-1* locus influence IL2 neural activity?

Again, we agree with your point that it would be interesting to pursue a mechanistic explanation of *nict-1* effect. For this specific experiment, we would like to measure IL2 neural activity using live calcium imaging in nictating animals. In separate projects in the Lee lab, we have attempted to set up calcium imaging for IL2 neurons in dauer stage for many years. However, we have been unsuccessful for several reasons. (1) Because nictation is a dynamic 3D behavior, it is extremely difficult to do live calcium imaging on freely behaving animals. (2)

We tried calcium imaging using microfluidic chambers in which dauer animals are trapped, but that developmental stage is too thin to capture effectively and we can not provide the appropriate mechanical stimuli to promote the behavior. We hope to overcome the technical issues in the future, but these experiments go well beyond what we believe is possible for this manuscript.

REVIEWERS' COMMENTS:

Reviewer #1 (Remarks to the Author):

Thankyou for addressing all the many comments and suggestions in a very positive manner.

Reviewer #3 (Remarks to the Author):

Lee et. al. describes a genetic locus that contributes to nictation behavior of *C. elegans* dauer larvae. After the revision, the authors address many reviewers' concerns and provide evidences and statistics that significantly enhance the claims of the authors. Therefore, I recommend the paper to be accepted for publication.